# A Highly Conserved Region in BRCA2 Suppresses the RAD51-Interaction Activity of BRC Repeats

**DOI:** 10.3390/vetsci10020145

**Published:** 2023-02-10

**Authors:** Zida Zhu, Taisuke Kitano, Masami Morimatsu, Kazuhiko Ochiai, Toshina Ishiguro-Oonuma, Kosuke Oosumi, Xianghui Lin, Koichi Orino, Yasunaga Yoshikawa

**Affiliations:** 1Laboratory of Veterinary Biochemistry, School of Veterinary Medicine, Kitasato University, Towada 034-8628, Japan; 2Laboratory of Laboratory Animal Science and Medicine, Graduate School of Veterinary Medicine, Hokkaido University, Sapporo 060-0818, Japan; 3Laboratory of Veterinary Hygiene, School of Veterinary Medicine, Nippon Veterinary and Life Science University, Tokyo 180-8602, Japan; 4Laboratory of Veterinary Physiology, Cooperative Department of Veterinary Medicine, Faculty of Agriculture, Iwate University, 3-18-8 Ueda, Morioka 020-8550, Japan

**Keywords:** BRCA2, RAD51, BRC repeats, highly conserved region, comparative molecular biology, mutation analysis, canine mammary tumor, human breast cancer

## Abstract

**Simple Summary:**

Mammary tumors are the most common tumor type in female dogs and one of the major problems in veterinary oncology. Breast cancer 2, early onset (*BRCA2*) mutations are associated with tumorigenesis in humans and dogs. BRCA2 is a DNA damage repair protein that interacts with RAD51, another DNA-damage-repair protein, to preserve DNA stability. Tumors, such as mammary tumors, are developed when their function or interaction is lost. It is currently unclear how the interaction activity of BRCA2 and RAD51 is regulated. We focused on a highly conserved region (HCR) near the BRC repeats, which are RAD51-interacting domains in BRCA2. We demonstrated that HCR regulated the RAD51-interaction activity of BRC repeats. In addition, we looked for HCR mutations in canine mammary tumors and discovered I1110M mutations. The effect of four HCR mutations, including I1110M, on the RAD51-interaction activity of BRC repeats was investigated. Three HCR mutations (A1108G, S1114P, and T1115P) found in canine mammary tumors or human breast cancers influenced the RAD51-interaction activity of BRC repeats. According to this study, HCR regulated the RAD51-interacting activity of BRC repeats, and HCR mutations may be related to tumorigenesis in both dogs and humans.

**Abstract:**

Mammary tumors are the most prevalent type of tumors in female dogs. Breast cancer 2, early onset (*BRCA2*) malignant mutations are associated with tumorigenesis in humans and dogs. BRCA2 plays a pivotal role in homologous recombination repair by recruiting RAD51 recombinase to DNA damage sites to maintain genome stability. To recruit RAD51, BRCA2 must interact with RAD51 via BRC repeats, but the regulation of this interaction has been unclear. In this study, we focused on a highly conserved region (HCR) near BRC repeats. Using co-immunoprecipitation and mammalian two-hybrid assay, we found that HCR suppressed the RAD51-interaction activity of BRC repeats and that substitutions of HCR phosphorylation sites affected it. In canine tumor samples, we found ten mutations, including a novel HCR mutation (I1110M) from canine tumor samples. The effect of four HCR mutations, including I1110M, on the RAD51-interaction activity of BRC repeats was tested. One of the HCR mutations found in canine mammary tumors increased the interaction, but the two mutations found in human breast cancers decreased it. This study suggested that the HCR regulated the RAD51-interacting activity of BRC repeats through HCR phosphorylation and that mutations in HCR may be related to tumorigenesis in both dogs and humans.

## 1. Introduction

Cancer is one of the leading causes of death in humans and dogs (*Canis lupus familiaris*) [1,2]. Mammary tumors are the most prevalent neoplasms in female dogs, with a high malignancy rate [3,4,5]. Breast cancer 2, early onset (*BRCA2*) is a tumor suppressor gene first identified in humans. BRCA2 mutations have been linked with increased susceptibility to breast cancer in human studies [6,7]. Consequently, germline BRCA2 mutations are associated with an increased risk of breast cancer. Given the high lifetime risk of breast cancer (81–88%) in women carrying this mutation, a genetic analysis, including the detection of deleterious mutations to identify carriers of the BRCA2 mutation, is recommended [6,7]. 

However, given the limited number of studies focusing on canine BRCA2, it has yet to be proven that BRCA2 mutation is associated with an increased risk of cancer. According to a single-nucleotide polymorphism study of intronic markers, the *BRCA2* gene locus has been associated with both malignant and benign mammary tumors [8,9]. Previously, we reported a loss of heterozygosity, which is one of the mechanisms of *BRCA2* inactivation in mammary tumors [10]. Reduced BRCA2 expression in canines is related to the development of mammary tumors [11]. These findings suggest that canine *BRCA2* mutations are likewise related to the initiation of tumorigenesis.

BRCA2 acts as a mediator of RAD51, a DNA recombinase required for homologous recombination (HR) in humans and, most likely, in dogs [12,13,14,15]. Human and canine BRCA2 have two domains to interact with RAD51: in the BRC repeats and the C-terminal RAD51 binding domain [16,17,18,19]. In humans and dogs, the BRC repeats are eight repeated and conserved motifs consisting of approximately 26 amino acids that play an important role in recruiting RAD51 to sites of DNA damage [12,16,17,20]. The RAD51 interaction of BRC repeats is important in HR. According to a previous study, the deubiquitination of RAD51 facilitates RAD51-BRCA2 binding and RAD51 recruitment, which are critical for proper HR in humans [21]. In addition to this finding, mutations in BRC repeats in human and canine BRCA2 have been reported in mammary gland tumors, implying a link to tumorigenesis (https://brcaexchange.org, accessed on 19 October 2021) [17,20,22]. However, we know very little about the mechanism that regulates the interaction between BRCA2 and RAD51.

There is a highly conserved region (HCR) in between BRC repeat 1 (BRC1) and BRC repeat 2 (BRC2). According to a recent study, three amino acid residues in human BRCA2 (S1106, S1124, and S1129) can be phosphorylated by DNA damage kinases, ataxia telangiectasia mutated (ATM), and ataxia and Rad3-related (ATR) [23]. This BRCA2 phosphorylation enhances the binding of protein phosphatase 2A (PP2A), one of the protein phosphatases. This interaction between BRCA2 and PP2A is required for HR-mediated DNA repair [23]. Although the RAD51-interaction activity of BRCA2 is not altered by PP2A interaction in their report, our previous study demonstrated that the region from BRC1 to BRC2 has stronger RAD51-interaction activity than the sum of single BRC1 and BRC2 domains [24]. This result indicated that the region between BRC1 and BRC2, where HCR is located, may regulate the interaction between the BRC repeats and RAD51.

In this study, we aim to reveal the function of HCR from a different perspective by examining the interaction between BRC repeats and RAD51, as well as the potential regulatory mechanism via phosphorylation. We also analyzed canine mammary tumor and tumor-free samples and identified a novel mutation (3538T>G (I1110M)) in HCR. This study sheds light on how the HCR participates in the RAD51-interaction activity of BRC repeats in BRCA2 and how mutations in HCR are also potentially related to tumorigenesis.

## 2. Materials and Methods

### 2.1. Multiple Sequence Alignment

To evaluate the conserved regions, canine BRCA2 (accession No.: NP_001006654.2), human BRCA2 (accession No.: NP_000050.3), feline BRCA2 (accession No.:NP_001009858.1), mouse BRCA2 (accession No.:NP_001074470.1), chicken BRCA2 (accession No.: NP_989607.3), sea turtle BRCA2 (accession No.: XP_037747982.1), frog BRCA2 (accession No.: ABP48763.1), gecko BRCA2 (accession No.: XP_015279682.1), and zebrafish BRCA2 (Accession No.: NP_001103864.2) amino acid sequences were obtained from the NCBI database (https://www.ncbi.nlm.nih.gov (accessed on 19 October 2021)). Multi-species alignments were performed using the multiple sequence alignment program Clustal Omega (https://www.ebi.ac.uk/Tools/msa/clustalo/ (accessed on 19 October 2021)) according to the instruction manual (www.clustal.org/download/clustalw_help.txt, www.clustal.org/download/clustalx_help.html (accessed on 19 October 2021)) and using default settings [25], and then visualized using ESPript (http://espript.ibcp.fr/ (accessed on 28 December 2022)) [26].

### 2.2. Cell Culture, Generate Cell Line, and Antibodies

HeLa cells were obtained from RIKEN Cell Bank and grown in Dulbecco’s modified Eagle’s medium supplemented with 10% fetal bovine serum. HeLa cells stably expressing the FLAG-HA-nuclear localization signal (NLS)-fused canine BRCA2 BRC repeats were generated.

Western blotting was performed using the following antibodies: anti-RAD51 (dilution 1:10,000, Bio Academia, Osaka, Japan), anti-PP2A-Cα/β (dilution 1:1000, Santa Cruz, CA, USA), anti-FLAG (2H8; dilution 1:1000, Trans Genic, Kobe, Japan), anti-FLAG (PM020; 1:1000, MBL, Nagoya, Japan), and anti-Lamin B1 (dilution 1:2000 MBL).

### 2.3. Co-Immunoprecipitation

Canine BRCA2 peptides stably expressing HeLa cells were washed with PBS, harvested by scraping, and centrifuged at 500× *g* for 5 min. The pelleted cells were resuspended with an equal volume of IP buffer 1 (20 mM Tris-HCl, pH 8.0; 50 mM KCl; 2 mM MgCl_2_; 0.5% Triton X-100; 10% glycerol; 1× protease inhibitor cocktail (Merck KGaA, Darmstadt, Germany); and 200 U/mL Benzonase) at 4 °C for 30 min to digest the genomic DNA. The resuspended cells were subsequently treated with IP buffer 2 (20 mM Tris-HCl, pH 8.0; 150 mM KCl; 2 mM MgCl_2_; 10% glycerol; and 1× protease inhibitor cocktail (Merck KGaA)) to extract the proteins. The extracts were centrifuged at 17,000× *g* and 4 °C for 10 min, and the supernatants were added with FLAG M2 affinity agarose (Merck KGaA) and were incubated with rotation at 4 °C overnight. The immunoprecipitants were recovered by centrifugation at 5000× *g* for 1 min and washed three times with IP buffer 2. The samples were eluted into a 50 μL of 1× lithium dodecyl sulfate (LDS) loading buffer (Thermo Fisher Scientific, Waltham, MA, USA) and 1× sample reducing agent (Thermo Fisher Scientific).

For the RAD51 immunoprecipitation, we used RAD51 polyclonal antibodies (2H8; Trans Genic) and Protein G Dynabeads (Thermo Fisher Scientific). Briefly, the supernatants of extracts were gently mixed at 4 °C overnight with 3 μg anti-RAD51. 20 μL Protein G Dynabeads were then added to the samples and incubated with rotation at 4 °C for 3 h to couple the immunocomplex to the beads. Dynabeads were recovered using a magnet stand, washed three times with IP buffer 2, and then eluted into 50 μL of 1× LDS buffer (Thermo Fisher Scientific) and 1× sample reducing agent (Thermo Fisher Scientific).

### 2.4. Mammalian Two-Hybrid Assay

The wild type and mutants indicated in each figure of cBRC1-HCR-BRC2 were cloned into the pM vector (TaKaRa, Siga, Japan) to generate fused peptides with Gal4 DNA-binding domain, which specifically bound to a plasmid containing Gal4 DNA-binding sites. Canine RAD51 (Accession No.: NM_001003043.1) was cloned into a pVP16 vector (TaKaRa) to produce a fused protein with VP16, a transcriptional activation domain, which directed RNA polymerase II to transcribe the gene. The pG5-Luc plasmid contained a firefly luciferase gene downstream of Gal4 binding sites and a TATA box. Once these plasmids were transfected, the interaction between the BRCA2 peptides and canine RAD51 resulted in the transcription of the firefly luciferase reporter gene. Therefore, a higher expression of firefly luciferase indicates a stronger interaction. In addition, pRL-TK (Promega, Madison, WI, USA) was also transfected to express the *Renilla* luciferase under the control of the SV40 promoter to normalize the transfection efficiency.

HeLa cells (5 × 10^4^ cells per well) were seeded in 24-well plates, with 50 ng of each pM-BRCA2 peptide plasmid, 50 ng of pVP16-RAD51 plasmid, 100 ng of pG5-Luc plasmid, and 10 ng of pRL-TK (Promega) plasmid transiently co-transfected using FuGENE HD transfect reagent (Promega) per well. Two days after transfection, the cells were washed with phosphate-buffered saline (PBS) and harvested using 1× passive lysis buffer (Promega). Firefly and *Renilla* luciferase activities were measured using a dual-luciferase reporter assay system (Promega) according to the manufacturer’s instructions and a luminometer (Luminescencer-PSN, AB-2200, ATTO, Tokyo, Japan).

### 2.5. Phosphorylation Assay by Phos-Tag SDS–PAGE

For the Phos-tag analysis of the HCR, cells were cultured in a 10 cm dish and X-ray irradiated using an MX-80Labo (mediXtec, Chiba, Japan) at a dose of 10 Gy. After irradiation, the cells were returned to the incubator for recovery. The cells were then harvested using cell scrapers and washed with Tris-buffered saline (TBS). The cell pellets were digested with an equal volume of Benzonase Buffer (20 mM Tris-HCl, pH 8.0; 50 mM KCl; 2 mM MgCl_2_; 0.5% Triton X-100; 10% glycerol; 1× PhosSTOP (Roche, Basel, Switzerland); and 200 U/mL Benzonase) at 4 °C for 10 min to digest the genomic DNA. The resuspended cells were subsequently treated with the same volume of 2% SDS to extract the total proteins. They were then mixed with 4× LDS loading buffer (Thermo Fisher Scientific) and 10× reducing agent (Thermo Fisher Scientific). The samples were electrophoresed using 10% acrylamide gel containing 100 µM Phos-tag (FUJIFILM Wako Pure Chemical Corporation, Tokyo, Japan) and 100 µM MnCl_2_ according to the manufacturer’s protocol and then immunoblotted using anti-FLAG antibody (2H8; dilution 1:500–2000, Trans Genic).

### 2.6. Tissue Sampling and DNA Isolation

In total, 81 blood samples from tumor-free dogs and 72 canine mammary tumor samples were obtained from the veterinary hospitals of Kitasato University, Iwate University, Hokkaido University, and Tokyo University of Agriculture and Technology or were kindly gifted by Dr. Kubo, Dr. Shimamura, and Dr. Heishima (Appendix A). Blood samples were stored at –20 °C until DNA extraction. Mammary tumor specimens were stored in RNAlater (Thermo Fisher Scientific) at –20 °C until DNA extraction. Genomic DNA was extracted from blood samples, mammary glands, or mammary tumor specimens using the Gentra Puregene kit (Qiagen, Hilden, Germany), according to the manufacturer’s instructions.

### 2.7. PCR and DNA Sequencing

PCR was performed using KOD FX Neo DNA polymerase (Toyobo, Osaka, Japan), according to the manufacturer’s instructions. Primer sets were listed in Appendix A. The PCR products were treated with endonuclease I (New England Biolabs, Ipswich, MA, USA) and shrimp alkaline phosphatase (USP, Rockville, MD, USA), and their base sequences were determined by direct sequencing.

### 2.8. Statistical Analysis

Data are presented as means, and statistical significance was set at *p* < 0.05. One-way ANOVA followed by Dunnett’s test or Tukey’s HSD test for multiple comparisons and the two-tailed Mann–Whitney *U* test were performed. Statistical analysis was conducted using JMP (SAS, Cary, NC, USA) and SPSS Statistics (IBM, New York, NY, USA). Graphs were constructed using Prism 9 (GraphPad Software, San Diego, CA, USA).

### 2.9. Sorting Intolerant from Tolerant (SIFT) Analysis

We used the “sorting intolerant from tolerant” (SIFT) process to evaluate the effect of missense mutations [27]. Mutations with a <0.05 SIFT score were determined as intolerant mutations. The SIFT program uses sequence homology to predict whether an amino acid mutation will affect protein function.

## 3. Results

### 3.1. Location and Alignment of the Highly Conserved Region in BRCA2

We aligned the BRCA2 amino acid sequence among animal species (Figure 1). The HCR is located in the center of BRC1 and BRC2 in mammalian, avian, and reptilian BRCA2 (Figure 1A). Its location was slightly different in zebrafish and *Xenopus*. The HCR in zebrafish BRCA2 is located at the N-terminus of BRC1 and is situated between BRC3 and BRC4 in *Xenopus*. Nevertheless, HCR conservation was higher than BRC1 or BRC2, which are critical BRCA2 domains (Figure 1B). Already reported phosphorylation sites and the PP2A binding motif in the HCR were also well conserved.

### 3.2. HCR Attenuates the Interaction Activity of BRC Repeat to RAD51

To determine whether the presence of HCR has an effect on the interaction between the BRC repeat and RAD51, stable cell lines expressing canine BRC1 or BRC2 with or without the HCR (cBRC1-HCR, 993–1139 aa; cHCR-BRC2, 1090–1248 aa; cBRC1-ΔHCR, 993–1100 aa; and cΔHCR-BRC2, 1134–1248 aa) were generated (Figure 2A,D). BRC1 expression levels were higher with HCR than without HCR. We also realized that exogenous BRC repeats altered the expression level of endogenous RAD51. Endogenous RAD51 was increased by expressing cBRC1-HCR or cBRC1-ΔHCR peptides (Figure 2A). Conversely, the expression of cHCR-BRC2 or cΔHCR-BRC2 resulted in a reduction in the expression of endogenous RAD51 (Figure 2D). We then assessed the RAD51-interaction activity by co-immunoprecipitation. The immunoprecipitated peptides of the cBRC1-ΔHCR had a lower yield than those of the cBRC1-HCR, but the yields of the co-immunoprecipitated endogenous RAD51 were comparable (Figure 2B). Thus, the RAD51 interaction of the cBRC1-ΔHCR was stronger than the cBRC1-HCR. The quantification result revealed that the cBRC1 peptide interacted more strongly with RAD51 without HCR than with HCR (Figure 2C). A similar result was obtained in cBRC2 (Figure 2E,F). The immunoprecipitated peptides of cΔHCR-BRC2 had the same intensity as that of cHCR-BRC2, and the yields of cΔHCR-BRC2-interacted RAD51 were higher than that of cHCR-BRC2 (Figure 2E). The quantification results showed that cΔHCR-BRC2 interacted more strongly with RAD51 than the cBRC2 peptide with the HCR (Figure 2F).

To further determine the effect of the HCR region more comprehensively, stable cell lines expressing canine BRCA2 peptides from BRC1 to BRC2 with the HCR (cBRC1-HCR-BRC2, 993–1248 aa) or without the HCR (cBRC1-ΔHCR-BRC2, 993–1248 aa deleted with 1101–1133 aa) were generated (Figure 3A). The expression level of endogenous RAD51 protein decreased in cells expressing cBRC1-HCR-BRC2 or cBRC1-ΔHCR-BRC2 which was consistent with the results obtained from HeLa cells expressing BRC2 (Figure 2E). We assessed the RAD51-interaction activity of cBRC1-HCR-BRC2 or cBRC1-ΔHCR-BRC2 by co-immunoprecipitation (Figure 3B,C). The immunoprecipitated peptides of the cBRC1-ΔHCR-BRC2 had a lower yield than those of the cBRC1-HCR-BRC2, but the yields of the coimmunoprecipitated RAD51 were comparable. Thus, without HCR, the RAD51 interaction of the BRC repeats was stronger than with the HCR. The relative quantification results normalized to the cBRC1-HCR-BRC2 revealed that the RAD51-interacting activity of BRC1 and BRC2 was significantly increased in the absence of HCR (Figure 3C). We also performed co-immunoprecipitation using anti-RAD51 polyclonal antibody to detect FLAG-tag fused peptides. To confirm the anti-FLAG co-immunoprecipitation assay, BRCA2 peptides and the RAD51 complex were immunoprecipitated using the anti-RAD51 antibody, and the peptides were detected via Western blotting. The co-immunoprecipitated peptide of cBRC1-ΔHCR-BRC2 exhibited approximately 1.5-fold stronger affinity for RAD51 compared to cBRC1-HCR-BRC2 (Figure 3D), which was consistent with the results obtained via immunoprecipitation using the anti-FLAG antibody (Figure 3B,C).

We also performed a mammalian two-hybrid assay to test the interaction and to confirm the co-immunoprecipitation assay. Consistently, cBRC1-ΔHCR-BRC2 interacted with RAD51 more strongly than cBRC1-HCR-BRC2 did (Figure 3E). The relative luciferase activity of the peptide possessing only HCR was the same level as that of cBRC1-HCR-BRC2 without the RAD51-expressing vector.

### 3.3. Mutations of Phosphorylation Sites in HCR Enhance RAD51 Interacting Activity of BRC Repeats

HCR possesses three sites that were shown to be phosphorylated by ATM and ATR in previous reports (Figure 4A) [23]. We hypothesized that phosphorylation might affect the RAD51 interaction activity of BRC repeats. First, we confirmed the phosphorylation of human HCR after X-ray irradiation using the Phos-tag SDS-PAGE (Figure 4B). Phosphorylated HCR from cells with and without irradiation was detected as a shifted band. HCR phosphorylation was increased 1 h after irradiation exposure. This suggests that the HCR is phosphorylated upon X-ray irradiation, which is consistent with previous reports [23].

In humans, three phosphorylation sites, S1106, S1123, and T1128, are phosphorylated by ATM/ATR [23]. To investigate the effect of phosphorylation in the HCR, we assessed the mutants of cBRC1-HCR-BRC2. Three phosphorylation sites in dogs, corresponding to S1106, S1123, and T1128 in human BRCA2, were substituted to alanine (S1105A, S1122A, T1127A, and 3A; S1105A, S1122A, and T1127A) or glutamic acid (S1105E, S1122E, T1127E, and 3E; S1105E, S1122E, and T1127E), constituting phosphorylation-disrupting or phosphorylation-mimicking substitutions (Figure 4A). Both phosphorylation-disrupting (3A) and phosphorylation-mimicking (3E) substitutions increased the interaction between BRC repeats and RAD51 compared with the cBRC1-HCR-BRC2. All mutations at individual phosphorylation sites, except S1122E, increased the RAD51-interaction activity of BRC repeats compared to the cBRC1-HCR-BRC2 (Figure 4C). HCR with S1105A only had a higher RAD51-interacting activity than HCR deleted peptide (cBRC1-ΔHCR-BRC2), and the other mutants had comparable or lower activity. We then compared the RAD51-interacting activity of the peptides between alanine and glutamic acid substitutions at the corresponding phosphorylation sites. The result was contradictory. The RAD51-interaction activity of 3E was greater than that of 3A when all three phosphorylation sites were substituted with glutamic acid. However, at S1105 and S1122, single glutamic acid substitutions had lower RAD51-interacting activity than that of alanine substitutions, while at T1127, both substitutions had similar activity.

### 3.4. Canine BRCA2 Mutations in the HCR in Mammary Tumor and Tumor-Free Samples

Because the HCR is highly conserved among animal species and regulates the RAD51-interaction activity of BRC repeats, we speculated that mutations in HCR might be associated with tumorigenesis in dogs. We determined the base sequence of canine *BRCA2* exon 11, which encodes the HCR and BRC repeats, in the mammary tumor samples (Table 1).

We identified 10 mutations in *BRCA2* exon 11 in 72 mammary tumor samples by comparing with the consensus sequence: 2135A>G (E643K), 2213A>G (N669D), 2329T>C (synonymous), 2609A>C (K801Q), 2696A>G (I830V), 3538T>G (I1110M), 4481A>C (T1425P), 4512A>G (K1435R), 5788G>A (synonymous), and 6886A>G (I2226M) (Table 1 and Appendix A). We also determined the base sequence in tumor-free samples. We found all mutations in 81 tumor-free blood samples (Table 1 and Appendix A). Homozygous mutations of 2135A>G (E643K), 2329T>C (synonymous), 3538T>G (I1110M), and 5788G>A (synonymous) were found only in mammary tumor samples (Table 1 and Appendix A). We analyzed the probability of deleterious mutations for non-synonymous mutations using SIFT score analysis, a score lower than 0.05 indicating intolerant mutations [27]. Mutations K801Q (2609A>C, SIFT score: 0), I1110M (3538T>G, SIFT score: 0), T1425P (4481A>C, SIFT score: 0), and I2226M (6886A>G, SIFT score: 0.05) were considered intolerant. The novel mutation I1110M (3538T>G, SIFT score: 0), predicted as an intolerant mutation, was located in the HCR, and homozygosity of this mutation was only found in mammary tumor samples.

### 3.5. The RAD51-Interaction Activity of BRC Repeats Was Altered by BRCA2 Mutations in the HCR Found in Mammary Tumors

We identified a novel mutation, I1110M, in the HCR. Then, we speculated that mutations in the HCR could contribute to tumorigenesis through BRCA2 dysfunction. Thus, we investigated the effect of mutations in the HCR on the RAD51-interaction activity of BRC1 and BRC2. Some mutations were also reported in HCRs from canine mammary tumors and human breast cancer samples (BRCA Exchange; https://brcaexchange.org (accessed on 21 July 2021)) [28]. The A1108G mutation in canine BRCA2 was previously found in the canine mammary tumor sample [28]. Almost every amino acid sequence in the HCR was reported in the mutations in the BRCA Exchange, a human BRCA2 mutation database. Here, we examined the effects of two canine mammary tumor mutations, I1110M and A1108G, as well as two additional mutations, S1114P and T1115P, which corresponded to S1115P (c.3343T>C) and T1116P (c.3346A>C) mutations from human breast cancer samples. These mutants were verified using SIFT score analysis; all four mutations were predicted as intolerant (SIFT score: 0). Consequently, we evaluated the effects of these mutations on the interaction of BRC1 and BRC2 with RAD51. Compared with the wild type (cBRC1-HCR-BRC2), A1108G increased the interaction activity, and no significant differences were found in the I1110M mutation. However, the S1114P and T1115P mutations reduced the interaction activity with RAD51 by mammalian two-hybrid assay (Figure 5A).

HeLa cells stably expressing canine cBRC1-HCR-BRC2 harboring A1108G, I1110M, S1114P, or T1115P mutants were generated to evaluate these mutations using co-immunoprecipitation. The expression of exogenous BRCA2 peptide and endogenous RAD51 or PP2A-C subunit (α and β) was assessed via Western blotting (Figure 5B). All cBRC1-HCR-BRC2 mutants, except the A1108G mutant, were detected. It was difficult to detect the signal in the A1108G mutant using whole-cell lysate, but there was a faint band with a small molecular weight band after immunoprecipitation (Figure 5C). In cells expressing cBRC1-HCR-BRC2 mutants, except the A1108G mutant, the expression level of endogenous RAD51 protein was decreased (Figure 5B). However, there was essentially no effect on the expression level of the PP2A-C subunit. Then, we evaluated the RAD51-interaction activity via co-immunoprecipitation (Figure 5C). Despite the low expression level of A1108G, the yield of immunoprecipitated RAD51 is higher than that of other BRCA2 peptides with or without mutations. The immunoprecipitated wild-type and mutant HCR peptides, including I1110M, S1114P, and T1115P, were essentially comparable. In S1114P and T1115P mutants, the yield of co-immunoprecipitated endogenous RAD51 was lower than in the wild-type but similar to the I1110M mutant. (Figure 5C). We also detected the interaction activity of these peptides with PP2A-C subunits. Surprisingly, all mutants, including cBRC1-ΔHCR-BRC2 peptides, interacted with the PP2A, and PP2A-C subunit-interaction activity of each mutant was stronger than that of wild-type peptide (Figure 5C).

## 4. Discussion

In this study, we identified the HCR as a RAD51-interaction regulatory region for BRC repeats located between BRC1 and BRC2 in BRCA2. The BRC repeats–RAD51 interaction is ordinarily attenuated by the presence of the HCR. A previous study found that the DNA-damage-signaling kinases ATM and ATR phosphorylated the HCR following DNA damage [23]. Consistent with this, we confirmed that the HCR was phosphorylated following X-ray irradiation. We then sought to determine the impact of phosphorylation on the RAD51-interacting activity of BRC repeats by phosphorylation-mimicking or -disrupting substitutions. Substituting ATM and ATR target sites in HCR influenced the RAD51-interaction activity of the BRC repeats. However, we could not obtain consistent results. Basically, alanine or glutamic acid substitutions at the ATM and ATR phosphorylation sites increased the RAD51-interacting activity of BRC repeats compared to the wild-type peptide. Furthermore, the sum of the RAD51-interacting activity of individual substitutions was not consistent with the RAD51-interacting activity of three amino acid residue substitutions. These phosphorylation sites were most likely structurally important, and substituting these amino acids resulted in the loss of HCR function, thus suppressing the RAD51-interacting activity of BRC repeats. Thus, we could not conclude how phosphorylation on HCR influenced the RAD51-interaction activity. However, HCR phosphorylation probably influences the RAD51 interaction because phosphorylation-mimicking substitution increases the RAD51 interaction activity.

It was reported that BRCA2 interacted with the phosphatase PP2A through the HCR and that the interaction between PP2A and HCR is critical for HR [23]. We also confirmed the interaction through co-immunoprecipitation. Although in the previous study, PP2A interacted only with the center of HCR, an HCR-deleted peptide (cBRC1-ΔHCR-BRC2) could still interact with PP2A. This result indicated that the other PP2A binding sites might be present outside of the HCR. Moreover, phosphorylation, as well as dephosphorylation, is important in controlling homologous recombination. It is worthwhile to investigate the PP2A phosphatase interacting sites outside HCR in the future.

It was also suggested that the HCR mutation disrupting the interaction with PP2A did not influence the RAD51-interaction activity of BRCA2 [23]. The PP2A-C subunit-interacting activities and the RAD51-interacting activity of BRC1 and BRC2 were not found to be connected. As a result, it was difficult to consider that HCR regulated the RAD51-interacting activity of BRC repeats through PP2A-interacting activity. The result suggested that HCR had at least two distinct functions. One is regulating the phosphorylation status of BRCA2 or BRCA2-interacting proteins through interacting with PP2A; the other is regulating the RAD51-interaction activity of BRC repeats through HCR phosphorylation, probably by ATM and ATR. Notably, we only used BRCA2 peptides and the impact of the HCR on full-length BRCA2 needs to be investigated in future studies.

According to our findings and those of others, the HCR plays an important role in BRCA2 function [23]. Mutations in the HCR are potentially associated with mammary tumors. As a result, we performed mutation analysis in canine *BRCA2* exon 11, which encodes the HCR and BRC repeats, from canine mammary tumor and tumor-free samples. We identified 10 mutations. The mutations 2135A>G (E643K), 2213A>G (N669D), 2609A>C (K801Q), 2696A>G (I830V), 4481A>C (T1425P), 4512A>G (K1435R), and 6886A>G (I2226M) have been previously reported by ourselves and other groups; however, the others were novel (2329T>C (synonymous); accession No. LC687404, 3538T>G (I1110M); accession No. LC687405, and 5788G>A (synonymous); accession No. LC687406) [10,15,28,29,30,31,32,33,34,35]. According to the SIFT scores, K801Q (2609A>C), I1110M (3538T>G), T1425P (4481A>C), and I2226M (6886A>G) were intolerant. In a recent study, the T1425P (4481A>C) mutation decreased the BRC repeat’s RAD51-interaction activity [17]. In this study, we did not perform any statistical comparison analysis because of the quite different group properties between mammary tumor and tumor-free samples. However, T1425P mutation might be associated with tumorigenesis. In our previous study, the K801Q (2609A>C) mutation was found to be a common polymorphism in dogs; thus, this mutation may be neutral even though the SIFT score suggested it was intolerant. Because the other two intolerant mutations, I1110M (3538T>G) and I2226M (6886A>G), were rare, we could not conclude them as malignant or neutral mutations. However, homozygosity of the I1110M (3538T>G) mutation was only found in mammary tumor samples; thus, it may be related to tumorigenesis. It is expected that large-scale case–control studies will elucidate the relationship between these mutations and mammary tumors in the future.

Because the I1110M (3538T>G) mutation is located in the HCR, we also examined the effect of mutations in HCR on the BRC repeats–RAD51 interaction activity. We also examined three other mutations: the A1108G mutation found in canine mammary tumor samples in the previous report, and S1114P and T1115P, corresponding to human BRCA2 S1115P and T1116P mutations, respectively, in breast cancer samples [28]. The peptide with A1108G mutation increased the interaction activity between the BRC repeats and RAD51 in mammalian two-hybrid assay and co-immunoprecipitation. This mutation disrupted the suppressive function of HCR. Notably, the relative value obtained from the co-immunoprecipitation was much higher than that from the mammalian two-hybrid assay (48-fold versus 1.5-fold). The expression level of peptides with A1108G mutation was extremely low, and the RAD51 expression level varied between the cells expressing wild-type peptides and the A1108G mutant. These factors might be related to the difference between co-immunoprecipitation and mammalian two-hybrid assay results. We also detected a small molecular weight band in the co-immunoprecipitation of peptides with the A1108G mutation. It suggested that the low expression level of the A1108G mutant is likely due to the cleavage and instability of the peptide. In conclusion, the A1108G mutation is likely to be malignant because of the loss of HCR suppression function and instability of the BRCA2 protein. The SIFT score suggested that the I1110M mutation is intolerant. Although the I1110M mutation did not influence the RAD51-interaction activity of BRC1 and BRC2, this mutation seems to increase the interaction activity with PP2A. However, additional studies are required to determine the impact of this mutation and to conclude on the effect on PP2A interaction or other BRCA2 functions and tumorigenesis. HCR with the other two mutations, S1114P and T1115P, reduced the interaction between BRC repeats and RAD51 but increased PP2A interaction. These mutations might change the three-dimensional structure of the HCR. They increase its inhibiting function for RAD51-interacting activity and its interacting function with PP2A. Human BRCA2 with the T1116P mutation has previously been reported to increase the sensitivity to the PARP inhibitor, inducing cell death in HR-deficient cells [23]. BRCA2 containing a T1116P mutation exhibits lower RAD51-interacting activity and possibly lower HR efficiency than wild-type BRCA2 and may increase PARP inhibitor sensitivity. In our study, we only used the BRC1-HCR-BRC2 peptides. Thus, to further investigate all four mutations, the effects of these mutations on full-length BRCA2 need to be explored in the future.

During the analysis of the RAD51-interacting activity, we realized that HeLa cells expressing the peptide containing BRC2 decreased endogenous RAD51 expression. This amount of reduced RAD51 expression seems to be related to the expression level of the peptide containing BRC2. We did not investigate how the BRC2 peptide decreased the expression level of endogenous RAD51, but this interesting issue should be elucidated in the future.

## 5. Conclusions

In this study, we identified a novel function of the HCR in BRCA2 located between BRC1 and BRC2 in canine BRCA2. The HCR regulated the RAD51-interaction activity of BRC repeats. We also confirmed the phosphorylation of the HCR, which could regulate the RAD51-interaction activity of BRC repeats. Additionally, we identified the novel homozygosity mutation, I1110M (3538T>G), in canine mammary tumor samples. Although this mutation did not have any effect on RAD51-interacting activity, the reported HCR mutation, A1108G from canine mammary tumors, increased the RAD51-interacting activity, and the two HCR mutations derived from human breast cancer, S1114P and T1115P, reduced the RAD51-interacting activity of BRC repeats. Thus, this study provides new insights into how BRCA2 regulates the interaction with RAD51 using the BRC repeats and the HCR, as well as offers possible future directions for research into the tumorigenic role of BRCA2 mutations in dogs and humans.

## Figures and Tables

**Figure 1 vetsci-10-00145-f001:**
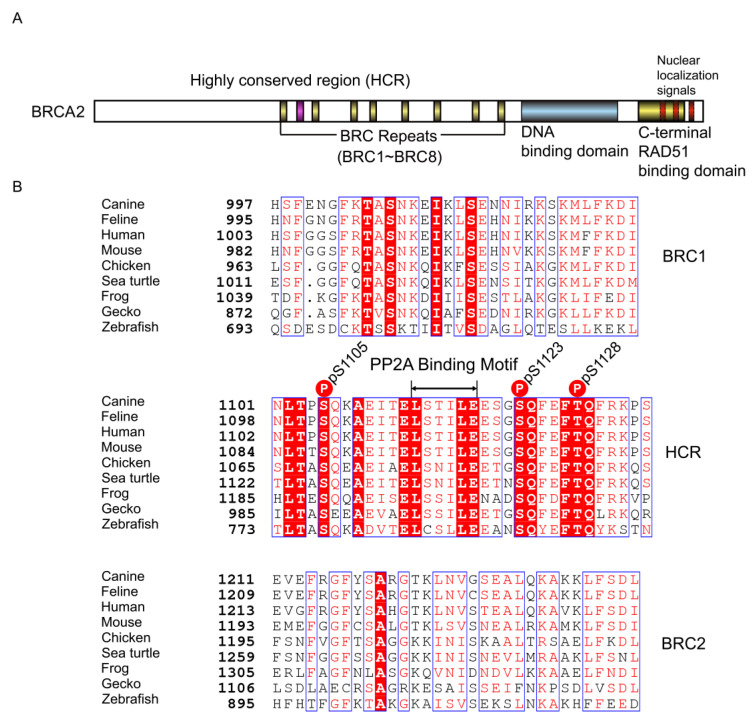
Location and alignment of the highly conserved region (HCR) between BRC repeat 1 (BRC1) and 2 (BRC2). (**A**) Schematic of the BRCA2 protein. Several major domains of BRCA2 were demonstrated, including the BRC repeats, DNA-binding domain, C-terminal RAD51-binding domain, and nuclear localization signals. BRCA2 interacts with RAD51 via the BRC repeats and C-terminal RAD51-binding domain. The HCR is located between BRC1 and BRC2 in canine BRCA2. (**B**) HCR, BRC1, and BRC2 alignment. Each canine, feline, human, mouse, chicken, sea turtle, frog, gecko, and zebrafish BRCA2 domain was aligned using ClustalW and ESPript (http://espript.ibcp.fr/, accessed on 19 October 2021). Identical residues are shown in red background; the conservative substitutions are grouped by blue boxes; red letters indicate similar residues according to the Risler matrix, and the remainder were shown in black [26]. The ATM and ATR phosphorylation sites and PP2A binding motif are marked in the figure [23]. Phosphorylation residue numbers are the amino acid numbers of the canine BRCA2.

**Figure 2 vetsci-10-00145-f002:**
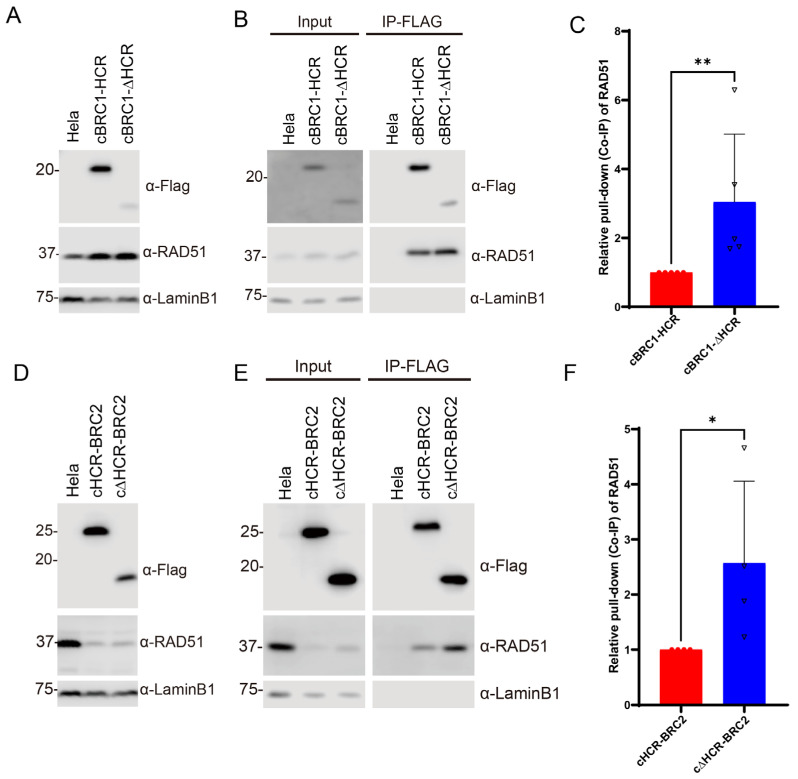
The highly conserved region (HCR) reduced the interactions between BRC repeat 1 (BRC1) or 2 (BRC2) and RAD51. (**A**) Expression of exogenous canine BRC1 peptide and endogenous RAD51. HeLa cells stably expressing FLAG-HA-tag and nuclear localization signal (NLS)-fused cBRC1-HCR (993–1139 aa) or cBRC1-ΔHCR (993–1100 aa) were determined via Western blotting using anti-FLAG and anti-RAD51 antibody, and anti-Lamin B1 antibodies (as loading control) (**B**) Co-immunoprecipitation assay to determine the interaction between cBRC1-HCR or cBRC1-ΔHCR peptides and endogenous RAD51. Lysates from HeLa cells or BRC1-peptide-expressing HeLa cells were immunoprecipitated with an anti-FLAG antibody. The immunoprecipitated samples were analyzed via Western blotting, as shown in (**A**). (**C**) Quantification of co-immunoprecipitated RAD51 with the BRC1 peptide in (**B**). Results were presented as the fold change compared with cBRC1-HCR. The data represent the mean ± SD of five independent experiments (*n* = 5) and circles and triangles represent the fold changes in each experiment. The Mann–Whitney *U* test was used to determine the significance of differences. Asterisks indicate significant differences (** *p* < 0.01). (**D**) Expression of exogenous canine BRC2 peptide and endogenous RAD51. HeLa cells stably expressing FLAG-HA-tag and NLS-fused cHCR-BRC2 (1090–1248 aa) or cΔHCR-BRC2 (1134–1248 aa) were determined via Western blotting as shown in (**A**). (**E**) The RAD51 interactions of cHCR-BRC2 and cΔHCR-BRC2 were compared using co-immunoprecipitation, as shown in (**B**). (**F**) Quantification of co-immunoprecipitated RAD51 with BRC2 peptide in (**E**). Results were presented as the fold change compared to cHCR-BRC2. The data represent the mean ± SD of four independent experiments (*n* = 4) and circles and triangles represent the fold changes in each experiment. The Mann–Whitney *U* test was used to determine the significance of differences. Asterisks indicate significant differences (* *p* < 0.05). The original western blot images are shown in Appendix A.

**Figure 3 vetsci-10-00145-f003:**
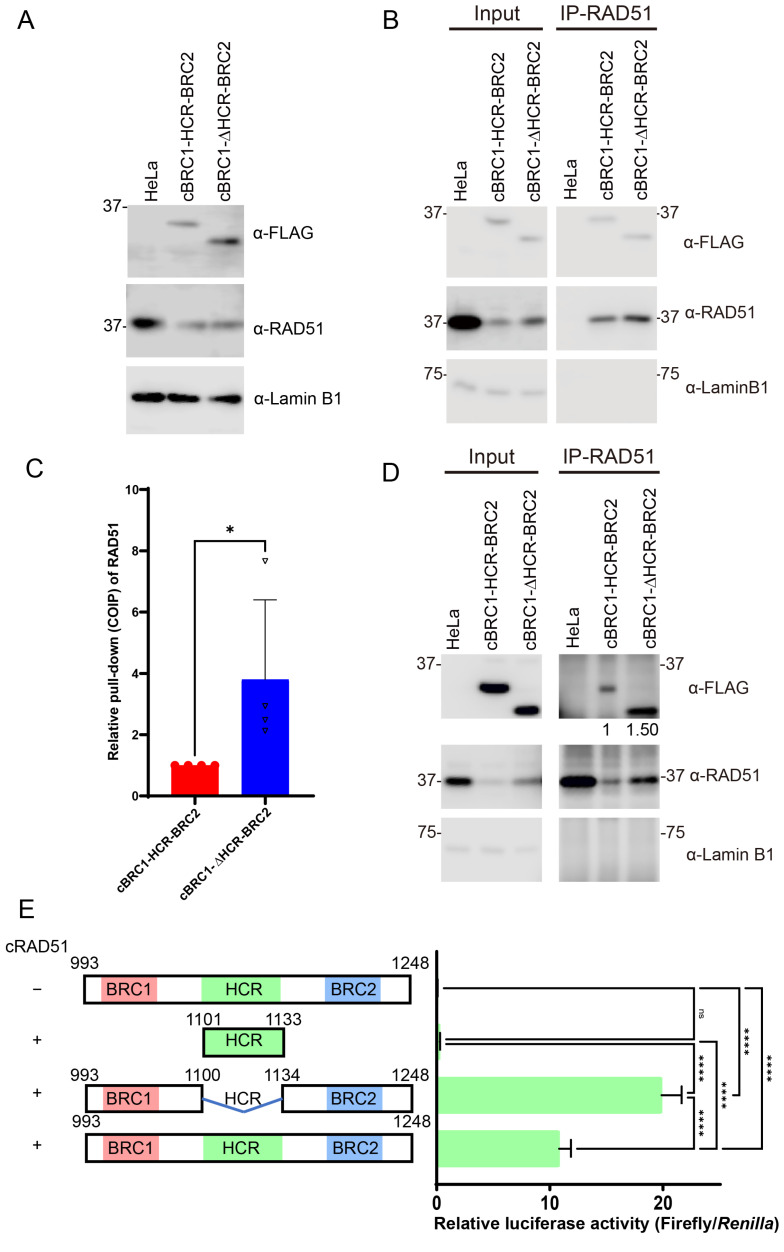
The highly conserved region (HCR) does not interact with RAD51 but reduces the interaction of BRC repeat 1 (BRC1) and 2 (BRC2) with RAD51. (**A**) Expression of exogenous BRCA2 peptides and endogenous RAD51. HeLa cells stably expressing FLAG-HA-tag and nuclear localization signal (NLS)-fused cBRC1-HCR-BRC2 and cBRC1-ΔHCR-BRC2 were determined via Western blotting using anti-FLAG and anti-RAD51 antibody, and anti-Lamin B1 antibodies (as loading control) (**B**) The RAD51 interactions of the cBRC1-HCR-BRC2 (993–1248 aa) and cBRC1-ΔHCR-BRC2 (993–1248 aa deleted with 1101–1133 aa) were compared using co-immunoprecipitation. The immunoprecipitated samples were analyzed via Western blotting using anti-FLAG, anti-RAD51, and anti-α-Lamin B1 antibodies. (**C**) Quantification of co-immunoprecipitated RAD51 with BRCA2 peptide. Results are presented as the fold change compared with cBRC1-HCR-BRC2; the data represent the mean ± SD of three independent experiments (*n* = 4) and circles and triangles represent the fold changes in each experiment. The Mann–Whitney *U* test was used to detect the significance of the differences. Asterisks indicate significant differences (* *p* < 0.05). (**D**) The RAD51-interaction activity between cBRC1-HCR-BRC2 and cBRC1-ΔHCR-BRC2 was compared via co-immunoprecipitation of RAD51. Immunoprecipitated samples were analyzed by Western blotting using anti-FLAG, anti-RAD51, and anti-Lamin B1 antibodies. The numbers under the anti-FLAG blot image indicate the relative values of co-immunoprecipitated BRCA2 peptides. (**E**) Interactions between RAD51 and cBRC1-HCR-BRC2 (993–1248 aa), cBRC1-ΔHCR-BRC2 (993–1248 aa deleted with 1101–1133 aa), and HCR (1101–1133 aa) were evaluated using a mammalian two-hybrid assay. An expression vector only transfected with cBRC1-HCR-BRC2 was used as the negative control. Results are presented as the mean, and error bars indicate the standard deviation (*n* = 4). Significance was examined using one-way ANOVA and Tukey’s multiple comparisons test. Asterisks indicate significant differences (**** *p* < 0.0001), and ”ns” indicates not significant. The original Western blot images are shown in Appendix A.

**Figure 4 vetsci-10-00145-f004:**
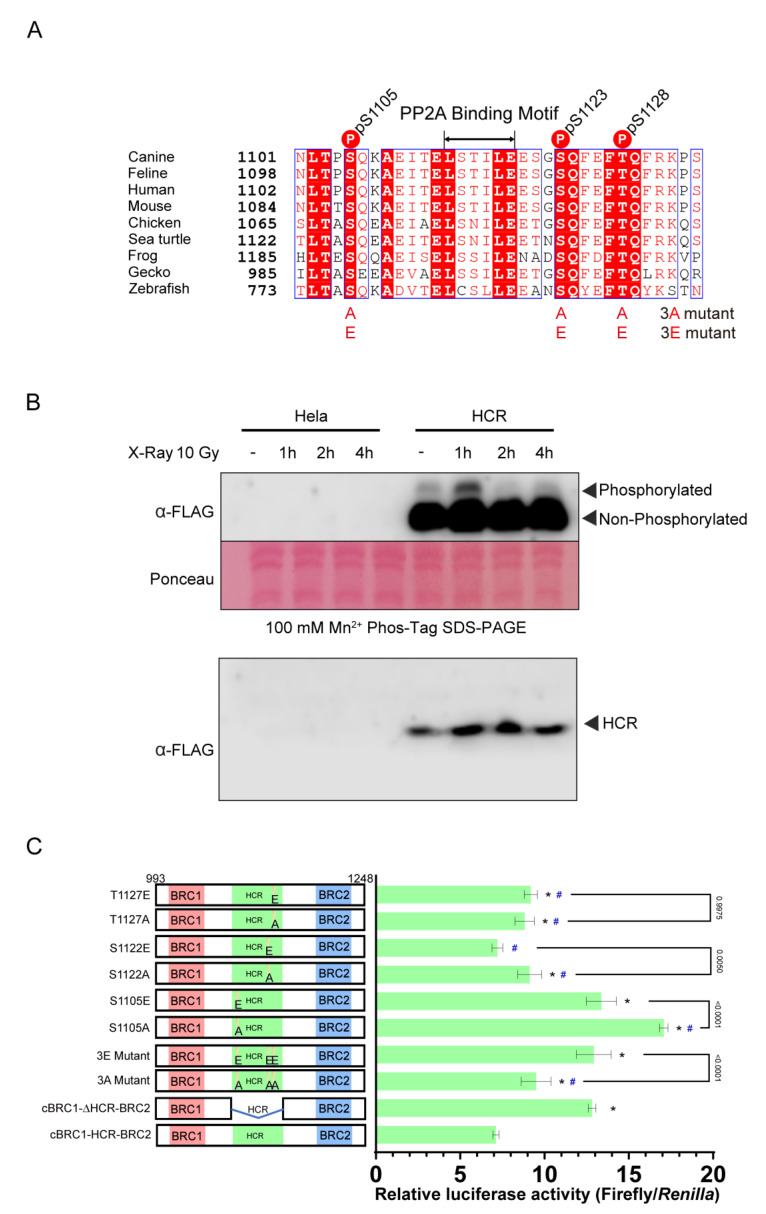
Phosphorylation of the highly conserved region (HCR) probably mediates the interaction between BRC repeats and RAD51. (**A**) Alignment of the HCR. Canine, feline, human, mouse, chicken, sea turtle, frog, gecko, and zebrafish HCRs in BRCA2 were aligned. Identical residues are shown in red background; the conservative substitutions are grouped by blue boxes; red letters indicate similar residues according to the Risler matrix, and the remainder were shown in black [26]. The sites of the alanine (3A; S1105A, S1122A, and T1127A) and glutamic acid (3E; S1105E, S1122E, and T1127E) substitutions are shown in the lower part of the diagram. (**B**) Phosphorylation in the HCR was detected via Phos-tag SDS-PAGE analysis. FLAG-HA-tag and nuclear localization signal (NLS)-fused human BRCA2 HCR, including peptides (1037–1191 aa), were stably expressed in HeLa cells. HeLa cells with or without HCR expression were irradiated with 10 Gy X-ray radiation and harvested at 1 h, 2 h, or 4 h post-irradiation. Lysates were analyzed through Phos-tag SDS-PAGE (top) and conventional 10% SDS-PAGE (bottom) at the same time using anti-FLAG antibodies. Phosphorylated proteins migrate slower due to their interaction with the Phos-tag reagent compared with unphosphorylated proteins. “Phosphorylated” and “non-phosphorylated” indicate phosphorylated and non-phosphorylated HCR, respectively. Ponceau staining was used to show equal loading. (**C**) The interaction activity between RAD51 and cBRC1-HCR-BRC2 (993–1248 aa) and its phosphorylation-mimicking (3E); non-phosphorylatable (3A); S1105A-, S1105E-, S1122A-, S1122E-, T1127A-, and T1127E-substituted peptides; and HCR-deleted peptide (cBRC1-ΔHCR-BRC2) was assessed using a mammalian two-hybrid assay. Results are presented as the mean, and error bars indicate the standard deviation (*n* = 4). Significance was examined by one-way ANOVA and Tukey’s multiple comparisons tests. Asterisks and sharps indicate significant differences from cBRC1-HCR-BRC2 (* *p* < 0.05) and from cBRC1-ΔHCR-BRC2 (# *p* < 0.05), respectively. The *p*-values with bars between alanine and glutamic acid substitutions at the corresponding phosphorylation sites are also shown. The original Western blot images are shown in Appendix A.

**Figure 5 vetsci-10-00145-f005:**
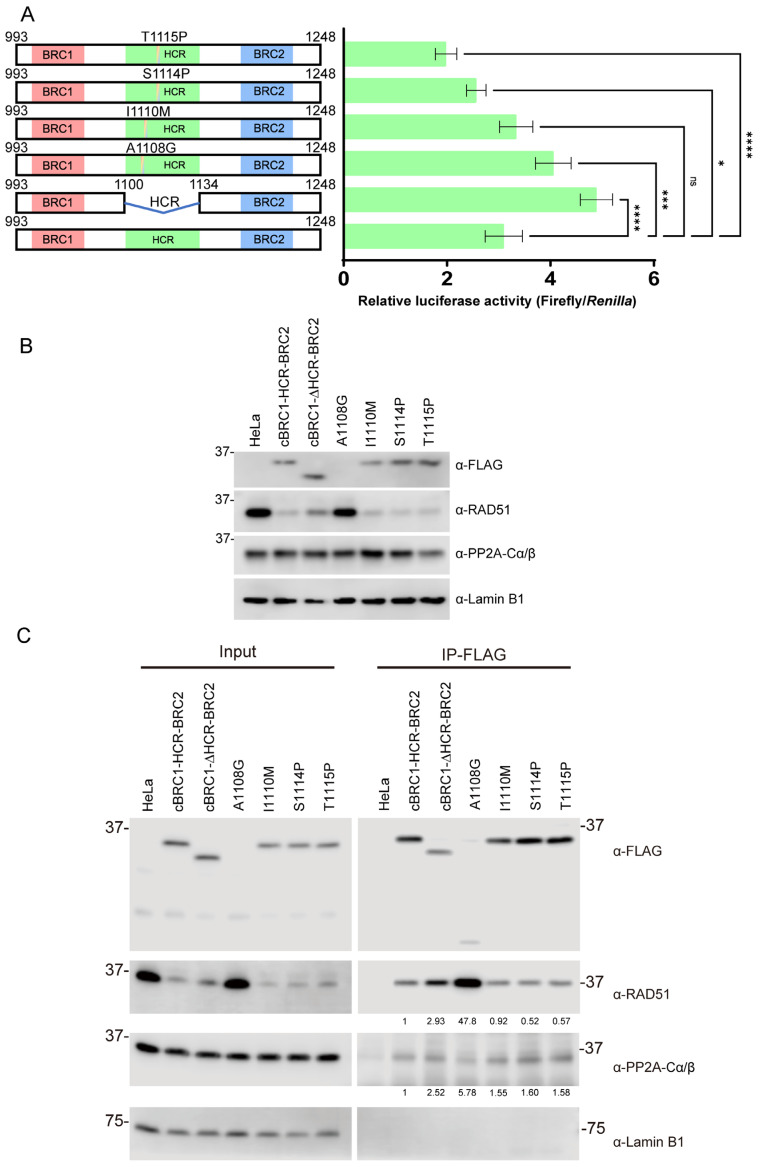
Mutations in the highly conserved region (HCR) identified in mammary tumor samples changed the RAD51-interaction activity of BRC repeats 1 (BRC1) and 2 (BRC2). (**A**) The interaction activity between RAD51 and cBRC1-HCR-BRC2, cBRC1-ΔHCR-BRC2, A1108G, I1110M, S1114P, or T1115P mutants was assessed using a mammalian two-hybrid assay. Results are presented as mean ± standard deviation (*n* = 4). Significance was examined using one-way ANOVA and Dunnett’s multiple comparison test. Asterisks indicate significant differences (**** *p* < 0.0001, *** *p* < 0.001, * *p* < 0.05), and ”ns” indicates not significant. (**B**) Expression of exogenous BRCA2 peptide and endogenous RAD51. HeLa cells stably expressing FLAG-HA-tag and nuclear localization signal (NLS)-fused cBRC1-HCR-BRC2, or its mutants were determined via Western blotting using anti-FLAG, anti-RAD51, anti-PP2A-C subunit (α and β), and anti-Lamin B1 antibodies (as loading control) (**C**) The RAD51-interacting activity of cBRC1-HCR-BRC2 with HCR mutations was assessed by co-immunoprecipitation. Lysates from HeLa cells expressing indicated peptides were immunoprecipitated with an anti-FLAG antibody. The immunoprecipitated samples were analyzed by Western blotting using anti-FLAG, anti-RAD51, anti-PP2A-C subunit (α and β), and anti-Lamin B1 antibodies (as loading control). The numbers under the images of RAD51 and PP2A-C subunit blots indicate the relative fold changes compared to cBRC1-HCR-BRC2 by quantification of co-immunoprecipitated RAD51 or PP2A with the BRCA2 peptide. The original Western blot images are shown in Appendix A.

**Table 1 vetsci-10-00145-t001:** Genotype frequencies and SIFT scores of mutations in tumor and tumor-free samples.

Variant		SIFT Score	Mutant Homozygosity		Heterozygosity		Remained Non-Mutant Sample	
Nucleotide Change	Amino Acid Change		Tumor Sample	Tumor-Free Sample	Tumor Sample	Tumor-Free Sample	Tumor Sample	Tumor-Free Sample
2135G>A	E643K	0.29	1	0	2	2	69	76
2213A>G	N669D	0.36	1	1	3	2	68	75
2329T>C	Synonymous		2	0	3	1	67	77
2609A>C	K801Q	0	13	24	23	20	36	36
2696A>G	I830V	0.59	1	1	3	2	67	78
3538T>G	I1110M	0	1	0	0	2	71	76
4481A>C	T1425P	0	5	1	2	8	64	72
4512A>G	K1435R	0.35	18	8	12	8	41	65
5788G>A	Synonymous		1	0	3	1	67	80
6886A>G	I2226M	0.05	1	1	4	2	66	78

Frequency of genotypes identified from canine mammary tumor-bearing and tumor-free dogs. The SIFT score ranging from 0 to 1 demonstrates whether changes in amino acid residues can influence the protein function (intolerant), and the amino acid mutation is predicted to be intolerant if the score is <0.05 [27].

## Data Availability

The datasets used and analyzed in the current study are available from the corresponding author upon reasonable request.

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
