# Peer review of "A Highly Conserved Region in BRCA2 Suppresses the RAD51-Interaction Activity of BRC Repeats"

_vetsci, 2023, doi:10.3390/vetsci10020145_

Round 1
Reviewer 1 Report
The authors reported that the HCR region regulated the RAD51-interaction activity of BRC repeats and tested several mutations’ effect on these binding sites, revealing the potential relevance to tumorigenesis in humans and dogs. Some of the evidence is not very convincing, so I’d like to see a major revision for this manuscript. Here’s some comments:
1. There’re some typing mistakes in the paper, please revise them and read carefully for the whole paper.
Line22: domines should be domain
Line193: biding should be binding
Line285: Figure 5A should be Figure 4A
Line365 and 369: should be Figure 5, no Figure 6 in this paper
Figure 3C, BRC1 showed a wrong label
In Figure 5B, I1110M showed a wrong label
The label of Figure 5C was not in a same size with Figure 5A and 5B
2. For Figure 1A, please label the C-terminal RAD51-binding domain.
3. For Figure 3A, the co-IP need a negative control, like Figure 2.
4. For Figure 4D, the experiment lacked a control of BRC1-2 without HCR.
5. Please give a possible explanation or discussion for why 3A/3E mutations increased the interaction between BRC repeats and RAD51.
6. For Figure 5A, the positive control of BRC1-2 without HCR was not obvious compared to Figure3C, so the result of A1108G is not convincing.
7. For Figure 5B, it needs a negative control, the sample of A1108G should be added to the assay, and the I1110M should be compared in a similar expression level with WT. Please change it.
Author Response
Reviewer #1:
The authors reported that the HCR region regulated the RAD51-interaction activity of BRC repeats and tested several mutations’ effect on these binding sites, revealing the potential relevance to tumorigenesis in humans and dogs. Some of the evidence is not very convincing, so I’d like to see a major revision for this manuscript. Here’s some comments:
Answer: We thank you for the critical comments and helpful suggestions. We have considered all these comments and suggestions.
- There’re some typing mistakes in the paper, please revise them and read carefully for the whole paper.
Answer: We appreciate you reading the article carefully and finding these errors. We apologize for not double-checking the details.
Line22: domines should be domain
Answer: Thanks for your insight and your suggestion. We have corrected this mistake accordingly (Line: 24).
Line193: biding should be binding
Answer: We have corrected this error accordingly (Line: 244).
Line285: Figure 5A should be Figure 4A
Answer: Thanks for your insight and your suggestion. We have corrected this mistake accordingly (Line: 388).
Line365 and 369: should be Figure 5, no Figure 6 in this paper
Answer: Thanks for your insight and your suggestion. We apologize for the confusion. We have corrected this mistake accordingly (Lines: 486 and 490).
Figure 3C, BRC1 showed a wrong label
Answer: Thanks for your insight and your suggestion. We have corrected this mistake accordingly (Figure 3E).
In Figure 5B, I1110M showed a wrong label
Answer: Thanks for your insight and your suggestion. We have corrected this mistake accordingly (Figure 5C).
The label of Figure 5C was not in a same size with Figure 5A and 5B
Answer: We have changed the font size of the label of Figure 5 accordingly (Figure 5).
- For Figure 1A, please label the C-terminal RAD51-binding domain.
Answer: Thanks for your insight and your suggestion. We have added the label of “C-terminal RAD51-binding domain” to the diagram accordingly (Figure 1A).
- For Figure 3A, the co-IP need a negative control, like Figure 2.
Answer: Thanks for your insight and your suggestion. We apologize for the error. We have now included a control group of HeLa in Figure 3 B (Figure 3B).
- For Figure 4D, the experiment lacked a control of BRC1-2 without HCR.
Answer: We appreciate this valuable suggestion. We have now integrated Figures 4C and 4D and generated a new Figure 4C. This experiment had several phosphorylation mutants with the control of BRC1-2 without HCR (Figure 4C).
- Please give a possible explanation or discussion for why 3A/3E mutations increased the interaction between BRC repeats and RAD51.
Answer: Thanks for your insight and your suggestion. That's a very interesting question. The results we obtained from phosphorylation-mimicking and phosphorylation-disrupting substitutions had a discrepancy. We could not explain how HCR phosphorylation influenced the RAD51-BRC repeats interaction. These phosphorylation sites are most likely crucial for HCR function. Thus, mutations at these sites resulted in the loss of HCR function. It was, therefore, difficult to conclude the result. After revising and changing, we described it in more detail in the discussion section (Lines: 547-568).
- For Figure 5A, the positive control of BRC1-2 without HCR was not obvious compared to Figure3C, so the result of A1108G is not convincing.
Answer: We are very grateful for your suggestions, and we have controlled the conditions more strictly to complete the experiment. The results revealed that A1108G was significantly different from the wild type, although it showed a lower increase in interaction compared to the control with BRC1-2 without HCR (Figure 5A). Now we can discuss the deleterious features of the A1108G mutation (Lines: 484-485, 619-631).
- For Figure 5B, it needs a negative control, the sample of A1108G should be added to the assay, and the I1110M should be compared in a similar expression level with WT. Please change it.
Answer: Thanks for your insight and your suggestion. We apologize for neglecting the negative control in the co-immunoprecipitation, and we repeated this experiment and included a negative control (Figure 5 C). Since it was very difficult to generate stable expression cells of A1108G due to the instability of the A1108G mutant, this mutation was not included in the figure. Now we produced this sample successfully and re-performed the immunoprecipitation. In the assay, the expression level of the BRCA2 peptides was similar except A1108G mutation (Figure 5 B).

Reviewer 2 Report
In this study, Zhu et al investigated the potential role of a highly conserved region (HCR) located between two BRC repeats of the breast cancer 2, early onset gene (BRCA2) in the regulation of BRCA2 interaction with the recombinase RAD51. They explored the consequence of HCR deletion in peptides carrying BRC repeats on RAD51 interaction by co-IP and mammalian two hybrid assays. They found mutations in canine and human breast cancer located in the HCR sequence and tested their impact on RAD51 interaction. Globally, the authors found that there is not much coherence in the effect of the mutations tested, which makes difficult to conclude about the role of the HCR. Additionally, the mammalian two hybrid results provided are convincing but the Co-Ips are not. The effect of the binding of PP2A to the HCR was not explored. A systematic mutagenesis of the most relevant residue of HCR could be explored.
Major revision:
Results:
L209 Co-Ip results in Fig 2A, 3A and 5B are not very convincing. The lower intensity of BRC1-∆HCR spot revealed by α-FLAG antibody might represent a lower affinity of the antibody for this peptide or a difficulty to detect it on the blot. Actually, the revelation of both peptides with α-FLAG antibody in extract lanes is random (Original data fig 2B). Fig5 B and C, Immunoprecipitated RAD51 signals follow very well the amount of RAD51 in the Input. This could indicate that RAD51 interaction is not affected.
To make sure of the interpretation made, an immunoprecipitation of RAD51 with RAD51 polyclonal antibodies and detection of the co-immunoprecipitated BRC peptides with an anti-FLAG antibody should be presented. FLAG should be written in uppercase. Is the sequence of the peptides expressed from canine or human origin? The higher signal for Rad51 observed in the control Hela cell in Fig 2C is not relevant. It seems that the amount of protein loaded or the culture conditions are not well controlled.
Because of the possible detection problem with peptides, Fig 2A and 3A experiments could be confirmed by mammalian two hybrid assay as in Fig3C.
The effect of the binding of PP2A should be explored.
A systematic mutagenesis of the most relevant residue of HCR could be explored.
Minor revisions:
Despite Editage help, another round of English editing is required.
Simple Summary:
L22 Rad51-intracting domAines in BRCA2
Introduction:
L50 According to WHO, cancer is not the primary cause of death in humans
L67 Citing ref 12 to 20, it should be specified if these studies are made in humans or dogs.
L72 Citation of ref 19 is misleading: deubiquitination of RAD51 leads to increase RAD51-BRCA2 binding and is thus required for proper Rad51 recruitment and thus is critical for proper HR.
L78 RAD51 is regulated.
L79 In between of BRC1 and 2
L84 Interaction between BRCA2 and PP2A is required for HR-mediated DNA repair
L86 this is in contradiction with the beginning of the paragraph that says that HCR is in the middle of BRC1 and 2. The basis of the question asked is not clear enough: what is the relationship between high homology and the alteration of the interaction between BRC and Rad51?
Materials and Methods:
L130 The mammalian two-hybrid assay should be better described in order to be quickly understood (for example, how luciferase is reconstituted and how is Renilla expressed?)
L144 precise what kind of radiation is delivered
Results:
L196 Figure 1: precise color code. Increase the police for pS and PP2A. put the domains in order, as in the protein sequence: BRC1, HCR, BRC2
L254 the title of Fig3 should be HCR does not interact with RAD51
L285 Fig4A
L286, fig4B is not relevant
L289, add reference to fig4C
L295 this increase is smaller
L299 HCR has already been defined. This title is not easy to understand: substitutions cannot mediate anything.
L320 the system of notation of significant differences with letters is not clear
L363 A1108G seems to increase Rad51 interaction as ∆HCR does. Error bars should be presented both ways. This mutation should be tested in co-Ip.
L365 Fig5
Author Response
Reviewer #2:
In this study, Zhu et al investigated the potential role of a highly conserved region (HCR) located between two BRC repeats of the breast cancer 2, early onset gene (BRCA2) in the regulation of BRCA2 interaction with the recombinase RAD51. They explored the consequence of HCR deletion in peptides carrying BRC repeats on RAD51 interaction by co-IP and mammalian two hybrid assays. They found mutations in canine and human breast cancer located in the HCR sequence and tested their impact on RAD51 interaction. Globally, the authors found that there is not much coherence in the effect of the mutations tested, which makes difficult to conclude about the role of the HCR. Additionally, the mammalian two hybrid results provided are convincing but the Co-Ips are not. The effect of the binding of PP2A to the HCR was not explored. A systematic mutagenesis of the most relevant residue of HCR could be explored.
Answer:
We would like to thank you for your thorough reading, useful comments and remarks, and constructive recommendations, all of which have significantly improved our manuscript.
We carefully considered all the reviewers’ comments and revised our manuscript accordingly. The manuscript has also been double-checked, and any typos or grammatical errors have been corrected.
Major revision:
Results:
L209 Co-Ip results in Fig 2A, 3A and 5B are not very convincing. The lower intensity of BRC1-∆HCR spot revealed by α-FLAG antibody might represent a lower affinity of the antibody for this peptide or a difficulty to detect it on the blot. Actually, the revelation of both peptides with α-FLAG antibody in extract lanes is random (Original data fig 2B). Fig5 B and C, Immunoprecipitated RAD51 signals follow very well the amount of RAD51 in the Input. This could indicate that RAD51 interaction is not affected.
Answer: Thank you for your suggestion. Another immunoprecipitation experiment with a control was conducted. We found that the immunoprecipitated HeLa sample contained almost no non-specific binding bands. We believe that the antibody binding was specific (Figure 3B, and Figure 5C). In addition, we obtained consistent results in all quantitative analyses, as detailed in “Original Image for quantitative analyses (Figure 2C and F, and Figure 3C).” Furthermore, in the PP2A co-immunoprecipitation, the T1115P-peptide co-immunoprecipitated 1.5-fold more PP2A than the wild type (Figure 5C). This result is consistent with the previous report (Ambjørn et al., 2021). Based on these facts, we consider the experiments reproducible. We would also like you to take note of the expression level of endogenous RAD51 (Figure 2D, Figure 3A, and Figure 5B). This expression level was reduced by expressing peptides containing the BRC repeat 2. This was another reason why loading the same amount of RAD51 was difficult.
To make sure of the interpretation made, an immunoprecipitation of RAD51 with RAD51 polyclonal antibodies and detection of the co-immunoprecipitated BRC peptides with an anti-FLAG antibody should be presented. FLAG should be written in uppercase. Is the sequence of the peptides expressed from canine or human origin? The higher signal for Rad51 observed in the control Hela cell in Fig 2C is not relevant. It seems that the amount of protein loaded or the culture conditions are not well controlled.
Answer: Thanks for your insight and your suggestion. We sincerely apologize for our mistakes. For both the main text and the figures, we changed the FLAG to upper case. Additionally, we labeled the peptide species we used. Despite our efforts to maintain optimal culture conditions, it appears that the cells inevitably experience a gradual decrease in BRCA2 peptide expression (Lines: 264-269, 309-314, and 494-500) . (Figure 2A and D, Figure 3A, and Figure 5B). This was most likely due to the toxic effect of BRC repeats. We appreciate your suggestion to use the RAD51 antibody for immunoprecipitation. The results of this experiment were generally consistent with the results of the anti-FLAG immunoprecipitation (Lines: 325-332) (Figure 3D). This greatly improves the reliability of the results.
Because of the possible detection problem with peptides, Fig 2A and 3A experiments could be confirmed by mammalian two hybrid assay as in Fig3C.
The effect of the binding of PP2A should be explored.
A systematic mutagenesis of the most relevant residue of HCR could be explored.
Answer: Thanks for your insight and your suggestion. We believe that the results are reliable because we have performed several immunoprecipitation experiments. It was difficult to complete the mammalian two-hybrid assay in such a short period of time. We regret that we were unable to complete the mammalian two-hybrid assay.
We investigated the impact of HCR mutations on the PP2A affinity (Lines: 492-510, 569-578) (Figure 5B and C). Five HCR mutations (deletion mutation, A1108G, I1110M, S1114P, and T1115P) exhibited an increased affinity for PP2A. We did not investigate further, but the result indicated that it would be worthwhile to explore in the future. We intend to investigate the impact of these relevant residues mutations on HCR by using full-length BRCA2 in future studies.
Minor revisions:
Despite Editage help, another round of English editing is required.
Answer: Thanks for your insight and your suggestion. We sincerely apologize for any inconvenience this has caused.We have requested another round of English editing.
Simple Summary:
L22 Rad51-intracting domAines in BRCA2
Answer: Thanks for your insight and your suggestion. We have corrected this according to the reviewer’s comment (Lines: 24).
Introduction:
L50 According to WHO, cancer is not the primary cause of death in humans
Answer: Thanks for your insight and your suggestion. We sincerely apologize for this error. We revised the wrong description (Lines: 57-58).
L67 Citing ref 12 to 20, it should be specified if these studies are made in humans or dogs.
Answer: Thanks for your insight and your suggestion. We have specified whether these studies were conducted in humans or dogs (Lines: 78-93).
L72 Citation of ref 19 is misleading: deubiquitination of RAD51 leads to increase RAD51-BRCA2 binding and is thus required for proper Rad51 recruitment and thus is critical for proper HR.
Answer: Thanks for your insight and your suggestion. We apologize for making this logical error. We have now resolved this issue (Lines: 86-88).
L78 RAD51 is regulated.
Answer: Thanks for your insight and your suggestion. We have now resolved the issue (Lines: 93).
L79 In between of BRC1 and 2
Answer: Thanks for your insight and your suggestion. We have resolved this issue according to this comment (Lines: 94).
L84 Interaction between BRCA2 and PP2A is required for HR-mediated DNA repair
Answer: Thanks for your insight and your suggestion. We have resolved this issue according to this comment (Lines: 100-101).
L86 this is in contradiction with the beginning of the paragraph that says that HCR is in the middle of BRC1 and 2. The basis of the question asked is not clear enough: what is the relationship between high homology and the alteration of the interaction between BRC and Rad51?
Answer: Thanks for your insight and your suggestion. We have revised the unclear description according to this comment (Lines: 101-107). The presence of the region between BRC repeat 1 and BRC repeat 2 increased the interaction between BRC repeats and RAD51 (Ochiai et al., 2011). Although it was previously reported that the RAD51-interaction activity of BRCA2 was not influenced by PP2A-interaction disrupting mutation, our previous result indicated that the region between BRC repeat 1 and BRC repeat 2, where HCR is located, was. Because the homology of HCR is so high, we speculated that HCR might regulate the RAD51-interacting activity.
Materials and Methods:
L130 The mammalian two-hybrid assay should be better described in order to be quickly understood (for example, how luciferase is reconstituted and how is Renilla expressed?)
Answer: Thank you for your suggestions. According to the reviewer’s comment, we revised the Method section and briefly introduced the principles of mammalian two-hybrid experiments and how Firefly luciferase and Renillaluciferase are expressed (Lines: 168-181).
L144 precise what kind of radiation is delivered
Answer: Thank you for your suggestion. It’s X-ray irradiated. We have added this annotation to the main text and Figure 3B (Lines: 192, 370-378, 420, 544).
Results:
L196 Figure 1: precise color code. Increase the police for pS and PP2A. put the domains in order, as in the protein sequence: BRC1, HCR, BRC2
Answer: Thanks for your suggestion. We are now presenting a precise color code for alignment now. The sequences are now in the order BRC1, HCR, and BRC2. We also modified the size of the labels in the figure (Figure 1A). We believe it is much clearer now.
L254 the title of Fig3 should be HCR does not interact with RAD51
Answer: We appreciate this suggestion; we have modified the title accordingly (Lines: 344).
L285 Fig4A
Answer: We have corrected the mistake accordingly (Lines: 38).
L286, fig4B is not relevant
Answer: Thanks for your insight and your suggestion. We sincerely apologize for this mistake. We have corrected this now (Lines: 388).
L289, add reference to fig4C
Answer: Thank you for this suggestion. We have added a reference to Figure 4C (Lines: 398).
L295 this increase is smaller
Answer: Thanks for your insight and your suggestion. It's true that “smaller” is more accurate. However, we have replaced the sentence in the revised manuscript (Lines: 392-407).
L299 HCR has already been defined. This title is not easy to understand: substitutions cannot mediate anything.
Answer: Thanks for this suggestion. Following the reviewer’s comment, we have revised the figure title (Lines: 412-413).
L320 the system of notation of significant differences with letters is not clear
Answer: Thanks for your suggestion. We have replaced with a more direct expression of significant differences (Figure 4C, Lines: 425-439).
L363 A1108G seems to increase Rad51 interaction as ∆HCR does. Error bars should be presented both ways. This mutation should be tested in co-Ip.
Answer: Thanks for your advice, we are very grateful for these valuable comments. We re-evaluated the mutation A1108G. Indeed, A1108G is most likely a malignant mutation due to the appearance of significant differences in Figure 5A of the mammalian two-hybrid assay. Now the error bars have presented both ways (Figure 3E, Figure 4C, and Figure 5A), and we added the A1108G mutant sample in co-Ip successfully (Figure 5B and C). Using immunoprecipitation assay, we obtained that the A1108G mutation increased the interaction of the BRCA2 peptide with RAD51(Line: 484-485, 618-630).
L365 Fig5
Answer: We are sorry for this mistake. We have corrected the mistake now (Line: 490).
Ambjørn, S. M., Duxin, J. P., Hertz, E. P. T., Nasa, I., Duro, J., Kruse, T., . . . Nilsson, J. (2021). A complex of BRCA2 and PP2A-B56 is required for DNA repair by homologous recombination. Nat Commun(1), 5748. doi:10.1038/s41467-021-26079-0 , pmid = 34593815 , pmcid = PMC8484605
Ochiai, K., Yoshikawa, Y., Oonuma, T., Tomioka, Y., Hashizume, K., & Morimatsu, M. (2011). Interactions between canine RAD51 and full length or truncated BRCA2 BRC repeats. The Veterinary Journal, 190(2), 293--295. doi:10.1016/j.tvjl.2010.11.001 , pmid = 21123097

Round 2
Reviewer 1 Report
The authors addressed my questions quite well.